# Assessment of Water, Sanitation, and Hygiene Conditions in Public Elementary Schools in Quetzaltenango, Guatemala, in the Context of the COVID-19 Pandemic

**DOI:** 10.3390/ijerph20206914

**Published:** 2023-10-13

**Authors:** Michelle M. Pieters, Natalie Fahsen, Christina Craig, Ramiro Quezada, Caroline Q. Pratt, Andrea Gomez, Travis W. Brown, Alexandra Kossik, Kelsey McDavid, Denisse Vega Ocasio, Matthew J. Lozier, Celia Cordón-Rosales

**Affiliations:** 1Center for Health Studies, Universidad del Valle de Guatemala, Guatemala City 01015, Guatemala; nfahsen@uvg.edu.gt (N.F.); rquezada@uvg.edu.gt (R.Q.); a.gomezbarillas@wsu.edu (A.G.); ccordon@uvg.edu.gt (C.C.-R.); 2National Center for Emerging and Zoonotic Infectious Disease, Centers for Disease Control and Prevention, Atlanta, GA 30329, USA; prl5@cdc.gov (C.C.); lue3@cdc.gov (T.W.B.); spj0@cdc.gov (A.K.); ngl7@cdc.gov (K.M.); rhq1@cdc.gov (D.V.O.); or mlozier@cdc.gov (M.J.L.); 3Epidemic Intelligence Service, Centers for Disease Control and Prevention, Atlanta, GA 30329, USA

**Keywords:** COVID-19, hand hygiene, students, primary schools, WASH infrastructure

## Abstract

Water, sanitation, and hygiene (WASH) services in schools are essential to reduce infectious disease transmission, including that of COVID-19. This study aimed to establish a baseline of WASH services in six public elementary schools in Guatemala, with a focus on hand hygiene. We used the WHO/UNICEF Joint Monitoring Programme (JMP) report indicators to assess the WASH infrastructure at each school. We collected water samples from easily accessible water points (pilas, or bathroom sinks) at each school to test for the presence of total coliforms and *E. coli*. In-depth interviews were carried out with teachers to understand hand hygiene practices and systems at school. Results indicate that all schools had water available at the time of the survey. All water samples at four schools tested positive for total coliforms and at one school, positive for *E. coli*. All schools had sanitation facilities, but services were limited. Only 43% of handwashing stations at schools had soap available. No school had disability-inclusive WASH services. Financial constraints and a lack of appropriate WASH infrastructure were the main barriers reported by teachers to meet hand hygiene needs at school. Appropriate access to WASH infrastructure and supplies could increase hand hygiene practices and improve learning conditions for students.

## 1. Introduction

Water, sanitation, and hygiene (WASH) services are necessary to maintaining good health [1]. Clean water and adequate sanitation help prevent waterborne diseases such as diarrhea, which can be life threatening, particularly for children [2]. Good hygiene practices, such as handwashing with soap and water, are highly effective at preventing the spread of infectious and gastrointestinal illnesses, including COVID-19 [3]. 

Adequate WASH services in schools are a precondition for creating a healthy environment for children. Since students spend the majority of their time at school, safe WASH services can improve their overall health by preventing waterborne and infectious diseases [4]. Additionally, WASH services can improve academic performance by reducing absenteeism due to illness and allowing students to focus on their studies without distractions related to hygiene concerns [5]. Access to clean and private sanitation facilities also play an important role in addressing gender disparities. Proper facilities enable girls to manage their periods properly and with dignity, without having to interrupt their education [6]. Accessible and inclusive WASH services (i.e., having support railings, obstacle-free spaces for wheelchairs, et cetera) can also promote equality for all students, regardless of their abilities, and fosters a supportive learning environment [7]. Access to WASH services in schools is a bridge between the educational institution and the broader community, fostering well-being in students that extends to their families and contributes to the overall health of the community [8].

Access to water and sanitation and quality education are specific targets of the United Nations’ Sustainable Development Goals (SDGs). Goals 4a, 6.1, and 6.2 collectively aim to create effective learning environments, ensure universal access to safe and affordable drinking water, and achieve adequate and equitable sanitation and hygiene for all [9]. In order to monitor the progress towards achieving the SDGs related to inclusive education and the availability of water and sanitation for all, the World Health Organization (WHO) and the United Nations Children’s Fund (UNICEF) established the Joint Monitoring Programme (JMP) for Water Supply, Sanitation, and Hygiene. Since 2018, the JMP produces reports on the status of WASH in schools [10]. These reports present indicators for monitoring drinking water, sanitation, and hygiene elements through service ladders, and provide valuable indicators by region and country for tracking progress. Additionally, JMP reports provide data and insights that can inform the development of policies and strategies to improve access to WASH at the national and local levels. These reports can also allow governments and donors to allocate resources more effectively and hold stakeholders accountable for their commitments to providing WASH services in schools [11].

The JMP 2000–2021 data update report on the progress concerning drinking water, sanitation, and hygiene services in schools evidenced that the world is not on track to achieve universal access to basic WASH in schools by 2030. In 2021, 71% of schools had basic drinking water services (improved source with water), 72% of schools had basic sanitation services (improved single-sex facilities that were usable), and 58% of schools had basic hygiene services (handwashing facilities with soap and water) [10]. Focusing on Latin America and the Caribbean, 74% of schools had basic sanitation services; however, unfortunately, this region did not have sufficient data to estimate basic drinking water and hygiene services in schools. Meanwhile, in Guatemala, the estimated proportion of schools with basic sanitation services was 76%, slightly higher than the regional average [10].

Despite apparent progress, low and middle-income countries, including Guatemala, continue to face challenges in ensuring the provision of safe WASH services in schools. These challenges often come from a combination of financial, infrastructure, and social factors [12]. Furthermore, with the COVID-19 pandemic, the closure of schools posed a risk to the continuity of education and the well-being of children. According to Guatemala’s Ministry of Health, over 1.2 million cases of COVID-19 were reported in the country, and of those cases, approximately 45,000 belonged to school-aged children (5–12 years old) [13]. 

While acknowledging the significance of WASH services in schools and the progress made, it is important to delve into the local context to understand the different challenges and opportunities for improvement. To our knowledge, there are no publicly available data on WASH infrastructure in schools in Guatemala (apart from sanitation estimates). Therefore, this study aimed to assess the current state of WASH services in six schools in Quetzaltenango, Guatemala with a specific emphasis on hand hygiene practices. The findings from this assessment will serve as a foundation for designing an intervention to improve WASH infrastructure and promote effective hand hygiene practices within the schools. 

## 2. Methods

### 2.1. Setting and Demographics

This study was developed by Universidad del Valle de Guatemala (UVG), with support from the US Centers for Disease Control and Prevention (CDC). It took place in six public elementary schools from three municipalities in the Quetzaltenango Department: San Miguel Sigüilá, San Juan Ostuncalco, and Concepción Chiquirichapa (Figure 1).

Twelve schools were selected by convenience based on UVG’s previous work experience in the area. Exploratory visits to all 12 schools were made to establish contact with the principals, present the project, and gauge interest in participating. Six schools met the inclusion criteria (open to in-person classes), and all six agreed to participate.

### 2.2. WASH Facility Assessment

The facility assessment evaluated the WASH-related infrastructure and practices in each school. The instrument was adapted from the JMP’s core questions and indicators for monitoring WASH in schools, tailoring the questions to the specific context and cultural nuances [14]. Principals were asked questions that ranged from basic school demographic information to questions regarding water availability, sanitation services, and hand hygiene resources. Data were collected and managed on tablets using Research Electronic Data Capture (REDCap), a secure, web-based software platform hosted at UVG [15,16]. Descriptive analysis of the data was performed using STATA version 17 (StataCorp LLC; College Station, TX, USA), and WASH services were categorized into the JMP service ladders. Based on these ladders, services are classified as basic, limited, and no service (Table 1) [11].

### 2.3. Water Quality Testing

Water was tested according to the WHO Guidelines for Drinking-water Quality. According to these guidelines, microbial contamination must not be detectable in a 100 mL sample, and the minimum residual concentration of free chlorine should be 0.2 mg/L [17]. Water samples were collected at each school and tested for free residual chlorine (FRC), total coliforms, and *E. coli*. Water samples were collected from easily accessible water access points, such as pilas (concrete sinks common in Guatemala) or faucets that served multiple purposes including handwashing, drinking, cooking, and cleaning. If a school had two different water sources (e.g., municipal and spring), samples were taken from access points corresponding to each source.

Before collecting the sample, water was run at full speed for 15 s to remove any stagnated water from the line. FRC levels were first tested on-site with the Hach Chlorine Color Disc Test Kit (Hach Company; Loveland, CO, USA) following the directions provided with the kit. For readings of <0.2 mg/L of FRC, a second, 100-mL sample was collected in a sterile container with sodium thiosulfate [18]. Collected samples were placed inside a Ziploc bag in a cooler with ice packs. The samples were kept under cold storage conditions while transported to the UVG laboratory and processed within 24 h of collection. Total coliforms and *E. coli* were enumerated using the IDEXX Colilert-18^®^ (IDEXX Laboratories Inc.; Westbrook, ME, USA) method [19].

### 2.4. In-Depth Interviews

The aim of the in-depth interviews was to understand the hand hygiene practices, supply management, and challenges in the schools. All interviews used a consistent guide comprised of questions about the responsibility of hygiene management and supplies, perceptions of hygiene before and during the pandemic, and the hand hygiene education students receive. Two trained research personnel carried out the interviews, with a convenience sample of two teachers at each school for a total of 12 interviews. Verbal consent was obtained prior to each interview, and interviews were conducted in Spanish. The interviews lasted approximately 30 min and were audio recorded with participants’ consent. Interview recordings were then transcribed verbatim and translated into English by two bilingual research members for analysis.

A codebook was developed using a combined inductive and deductive approach; initial codes were developed based on interview questions and topics, and additional codes were added as themes emerged in the transcripts. Two reviewers independently coded three transcripts using MAXQDA 2020 software (VERBI Software, Berlin, Germany) and met to discuss consensus on code application and definition; later, one researcher applied the revised codebook to all other transcripts [20]. After all the transcripts were coded, one researcher used a thematic analysis approach to write analytic memos summarizing key themes across the interviews. Findings from the in-depth interviews were categorized under four themes: WASH-related education, hand hygiene supply management, hand hygiene practices, and WASH challenges and recommendations.

## 3. Results

### 3.1. General Characteristics and School Population

Of the six schools enrolled in the study, two were classified as urban (Schools 1 and 3) and four as rural (Schools 2, 4, 5, and 6). The total number of enrolled students at all schools was 2222, with an average of 370 (range 143–604, standard deviation, 183) students per school. On average, 55% of students were boys, and 45% were girls. Because of the COVID-19 pandemic, all schools were operating at limited student capacity, meaning students were divided into two approximately equal groups, which attended in-person classes on alternate days (a “bubble” model).

### 3.2. Water: Availability, Quality, and Accessibility

Results from the facility assessment demonstrate that all schools had water available from an improved source (piped water supply and/or protected spring) at the time of the survey, and the water access point was located inside the school grounds. Schools 1, 3, and 5 receive their water supply from the local municipality, and Schools 2, 4, and 6 receive water from a community-managed source. There is no disinfection system in place at the school level, and it is unclear if the municipal or community-managed water systems are equipped with chlorine disinfection systems. Five schools (all except School 3) had elevated water storage tanks. In addition to having elevated tanks, four schools (Schools 1, 4, 5, and 6) used small containers or plastic barrels to store water. Two schools (Schools 1 and 6) reported not having water for at least one day during the last academic year. Reasons for the reported water availability issues were mechanical problems with the pump or seasonal shortages.

Eight water samples were collected, as two schools had two different water sources. No school had detectable chlorine levels in the water (Table 2). All the samples from four schools (67%) tested positive for total coliforms, and one of these (17%) tested positive for *E. coli*.

### 3.3. Sanitation: Availability, Functionality, and Accessibility

All schools had flush or pour flush toilets inside the school. Three schools (Schools 2, 4, and 6) had single-sex facilities, and none of these had functioning infrastructure (e.g., lock did not work, toilets were broken). The student-to-toilet ratio across all schools averaged 44:1 at normal student capacity and 18:1 under the bubble model. At the time of the survey, no school had sanitation facilities accessible by students with physical disabilities, and only one (School 5) had sanitation facilities accessible by smaller children. 

### 3.4. Hand Hygiene: Availability and Accessibility

The mean number of handwashing (HW) stations per school was 11. Of these, 43% of stations had soap available at the time of the survey (Table 3). Under the bubble model, the student-to-HW station ratio was 32:1, and at normal student capacity, the ratio was 79:1. All schools had a HW station within 5 m of the sanitation facilities. No school had accessible HW facilities for students with physical disabilities, and two schools (Schools 5 and 6) had facilities for smaller children. Schools had an average of 14 dispensers of alcohol-based hand rub (ABHR) (range = 8–19). The ABHR dispensers were mostly stationed in classrooms or at classroom entrances and were also located in the principal’s office and at school entrances.

### 3.5. JMP WASH in Schools Service Ladder Framework

We classified the schools by the categories of the JMP service ladder framework (see Appendix A). We found that all schools had water available from an improved source at the time of the survey, but no school provided water with adequate chlorine residuals to meet Guatemalan drinking water standards. However, because schools did have water available from an improved source, they are categorized as having basic service. Although all six schools had improved sanitation facilities, the lack of functioning infrastructure placed them in the category of limited sanitation service. All schools were categorized as having basic service because they had at least one station with water and soap available at the time of the survey, except for School 5, which was classified as having limited service.

### 3.6. In-Depth Interviews

Twelve teachers participated in the in-depth interviews about water, sanitation, and hand hygiene rules and practices at schools.

#### 3.6.1. WASH-Related Education

All interview participants stated that students receive hygiene education, including hand hygiene, both through the National Curriculum, a document set by the Ministry of Education (MINEDUC) that guides what competencies a student should develop in each grade, and through informal lectures on teachers’ own initiative. Curriculum includes topics of personal hygiene in general, including the correct ways of handwashing. Though the National Curriculum includes modules designated for hygiene, it does not provide specific activities or materials to teach the modules. Some teachers reported having observed that, since the onset of COVID-19, students perform hand hygiene more regularly, regardless of whether it is discussed in the classroom.


*Teacher 0202: In my [class], I explain to the children how they should wash their hands and that it is very important to do it daily. And well, the children … are already doing it. They do it, mostly because of the circumstances [COVID-19 pandemic].*


#### 3.6.2. Hand Hygiene Supply Management

Participants mentioned three main ways that schools source hand hygiene resources for their students: (1) a hygiene-specific fund from MINEDUC, (2) teachers’ own resources; or (3) students’ and parents’ contributions. Most hand hygiene supplies come from the “fondo de gratuidad”, a fund provided to the schools twice a year specifically assigned from MINEDUC to purchase hygiene supplies. This fund is received by the school but managed by a Parent Organization Committee, which sources and purchases the supplies. Teachers are responsible for ensuring they have enough stock of supplies in their own classrooms, even if the school cannot provide the resources. Many teachers stated/reported/said that receiving funds only twice a year is insufficient to meet the students’ needs, forcing them to ration supplies so that they last longer. Since most supplies run out before the next installment arrives, teachers often purchase supplies with personal funds. If teachers cannot provide supplies on their own, they ask parents to send bottles of soap or ABHR for their children’s classrooms.

#### 3.6.3. Hand Hygiene Practices

Interviewees reported that hand hygiene practices at school have changed because of the pandemic. Interviewees noted that now, both teachers and students use ABHR and that teachers instruct students to wash their hands more often. Participants also stated that students are more aware of the importance of hand hygiene and choose to wash or clean their hands frequently.


*Interviewer: What changes, if any, have you noticed in school staff regarding their handwashing attitudes since the pandemic began?*



*Teacher 0102: Now every teacher uses gel [ABHR] before entering the principal’s office. Before [the pandemic] this was not done. No gel [ABHR] was applied.*



*Interviewer: And with your students? Have you observed any changes with respect to their attitudes about hand hygiene?*



*Teacher 0102: Yes, as I was saying, they have been more responsible about washing their hands, coming [to school] clean, and applying gel [ABHR].*


Regarding differences in hand hygiene practices between ages and genders, teachers mentioned that younger students wash their hands more often than older students since they are instructed to do so. However, younger children appear to have a harder time following proper hand hygiene steps. Several participants mentioned that girls are more aware of cleanliness, actively encouraging boys to wash their hands.


*Interviewer: Do you notice a difference between boys and girls or are they the same, in that sense [in hand hygiene practices]?*



*Teacher 0801: Yes, there is always some difference. The ones that get more clean are the girls.*



*Interviewer: The girls … Ok.*



*Teacher 0801: The boys are the ones that struggle a little bit.*


#### 3.6.4. WASH Challenges and Recommendations

The main challenges to proper hand hygiene mentioned by interview participants were lack of supplies and WASH infrastructure. In some schools, the existing infrastructure is not adequate for the number of students. Two participants from different schools mentioned that their institution has all hand hygiene stations centralized in one space. This caused problems when teachers wanted to avoid overcrowding at stations, especially in the context of COVID-19, when physical distancing was prioritized. Another challenge was water scarcity. Teachers reported that schools often run out of water, or only receive water for a limited time period.


*Interviewer: What are some of the things that you’ve observed that make it difficult for students to wash their hands or clean their hands?*



*Teacher 0601: Difficulty for us: Water. Water and handwashing stations … we can’t send a whole class to wash [their hands] because we don’t have water … we have water every other day …*


Participants were asked about how to improve hand hygiene practices at their schools. Some recommendations included increasing the number of handwashing stations with soap and water and distributing them throughout the school. Some teachers also mentioned that having size-appropriate handwashing stations for smaller children was necessary, as the current infrastructure is only suitable for taller children. Other suggestions included having murals, or painting the handwashing stations to attract students to those areas. If educational trainings were to be carried out, teachers recommend that the activities be hands-on and engaging for the students. Participants suggested the use of a skit with the proper steps for handwashing, or the implementation of a “Handwashing Festival” with activities where students could play and learn at the same time.

## 4. Discussion

To our knowledge, there are no public data on drinking water and hygiene infrastructure indicators in schools in Guatemala. Monitoring these indicators can aid local and national stakeholders in identifying and addressing gaps in WASH services and improve water quality, sanitation facilities, and hygiene practices in educational institutions. The current study was based on a convenience sample of six schools in Quetzaltenango that revealed that WASH infrastructure and the acquisition and management of hand hygiene supplies in these schools could be improved.

Although all schools in the sample had water available at the time of the survey, water scarcity issues remain a challenge for some schools. Water was not adequately disinfected to meet Guatemalan drinking water standards established by Decree Number 12–2002, which mandates that municipalities are responsible for providing a supply of potable, duly chlorinated water [21]. It is unclear why, or if, the municipal or community-managed water systems are not equipped with chlorine disinfection systems. Historically, there have been cultural and political factors that have created a distrust of the water supply, causing leaders to avoid chlorinating water [22]. Additionally, the health effects of handwashing with water contaminated with *E. coli* are still unknown, but evidence suggests that it might increase the risk of diarrheal diseases [23]. Increasing storage capacity and improving water quality could improve students’, teachers’, and the community’s health, especially because schools do not provide alternative drinking water sources.

The JMP service ladder is a useful framework to measure and track progress indicators related to WASH services in relation to the SDGs. However, it does not provide a definition of drinking water. Data to fulfill the indicators established by the JMP are observational only, and water quality cannot be determined by observation alone. Therefore, in our study, we considered schools to have basic service of water if they had water available from an improved source at the time of the survey, regardless of water quality test results.

All the schools in our sample had toilets in the facilities; however, most of these restrooms lacked the appropriate sanitation conditions (e.g., locks that work, privacy). We did not collect data on the number of toilets per gender bathroom (in schools that had single-sex facilities); therefore, we cannot calculate an adequate student-to-toilet ratio and whether these comply with WHO/UNICEF standards. The overall ratio of 18:1 under the bubble model was closer to WHO standards, while that of schools operating at full attendance is not (44:1). This indicates that there is a need to increase the number of sanitation services at the schools. WHO standards specify one toilet per 25 girls and one toilet plus a urinal for every 50 boys [24].

In their June 2020 COVID-19 Interim Guidance, the WHO explicitly recommended that all community locations (which includes schools) have water and soap available within 5 m of all toilets to increase handwashing practices [25]. Prior to the COVID-19 pandemic, these guidelines were only recommended for health-care facilities. Although all of schools in our study complied with this guidance, the ratio of students to hand hygiene stations is high, and teachers reported difficulty having all students wash their hands at designated times (32:1 under bubble model, 79:1 at full capacity). Currently, there is no standard ratio for HW stations to students.

Functional access to WASH facilities for people with disabilities in schools is critical to achieving a safe and inclusive environment for all, as proposed by the SDGs. None of the schools in our study had disability-inclusive WASH facilities [6]. According to the United Nations Convention on the Rights of Persons with Disabilities, an unaccommodated environment impedes students’ full participation in society and therefore places them on an unequal basis with others around them [6].

According to the in-depth interviews, schools faced financial constraints with regard to purchasing hand hygiene supplies and improving WASH infrastructure, leading to a lack of sufficient resources and the absence of adequate facilities to meet the students’ needs. The MINEDUC provides schools with 1.03 USD per student per school, per year (divided into two installments through the fondo de gratuidad), which seems insufficient; however, economic studies should verify this assumption [26]. A study carried out in Kenyan primary schools collected data from NGOs, government offices, and shops to calculate current expenditures on WASH and estimated costs for bringing the schools up to basic WASH standards [27]. Their calculations indicated that schools need an average of 3.03 USD per student, per year to keep up with recurrent WASH costs in schools with existing WASH infrastructure. A similar economic analysis of WASH costs in Guatemalan schools would be beneficial for MINEDUC and policy makers.

A strength of this study was that we were able to collect both quantitative and qualitative data to assess a more comprehensive understanding of WASH services in schools. Additionally, as recommended by existing literature, conducting 12 interviews ensured a high level of thematic saturation (more than 80%) [28]. Due to COVID-19 restrictions, we were only able to work with six schools, which can limit the generalizability of the results. Since the schools were selected by convenience, our sample might not be representative of the larger Guatemalan population or other settings. However, nonsystematic observations of other schools indicate that the conditions we describe are common. More extensive and systematic surveys are needed to document conditions and identify interventions to improve WASH infrastructure. As data for this study was only collected once, the information in this study does not capture changes or trends over time that could have changed.

## 5. Conclusions

Challenges schools face regarding access to safe WASH services are related to a lack of clean drinking water, sufficient sanitation resources (toilets), and hygiene supplies (soap and clean water). Teachers and families face much of the burden associated with the lack of WASH supplies, because if the school is not able to provide the necessary supplies, teachers often spend their own money on them. Now, more than ever, it is necessary to address the gaps related to WASH in schools to prevent this impact from having serious consequences for societal development and to be prepared for the next outbreak. The data collected from this study will be used to develop interventions to improve handwashing at the participating schools. Furthermore, policymakers and educational authorities can use this data as evidence of the need to allocate more resources to WASH services in schools. These changes and investments can enhance the quality of learning and the health of children and adolescents, reduce infectious diseases, and fulfill SDG Goals 4 and 6 of ensuring effective learning environments and access to water and sanitation for all.

## Figures and Tables

**Figure 1 ijerph-20-06914-f001:**
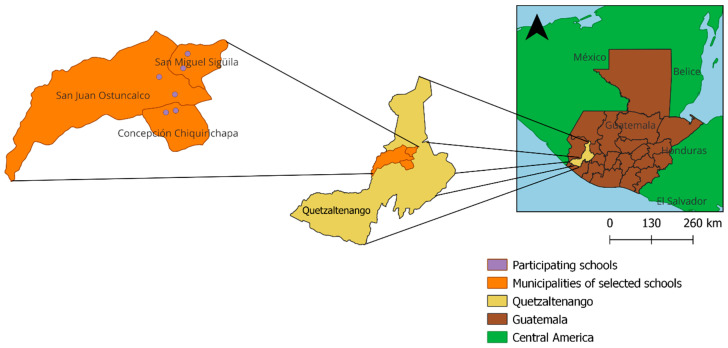
Map of Guatemala showing study locations.

**Table 1 ijerph-20-06914-t001:** JMP service ladders for monitoring drinking water, sanitation, and hygiene (WASH) in schools [11].

Service Level	Drinking Water	Sanitation	Hygiene
Basic service	Drinking water from an improved source and water that is available at the time of the survey	Improved sanitation facilitiesat the school that are single-sex and usable (available, functional, and private) at the time of the survey	Handwashing facilities with water and soap that is available at the school at the time of the survey
Limited service	Drinking water from an improved source but water that is unavailable at school at the time of the survey	Improved sanitation facilities at the school that are either not single-sex or not usable at the time of service	Handwashing facilities with water but no soap available at the school at the time of the survey
No service	Drinking water from an unimproved source or no water source at the school	Unimproved sanitation facilities or no sanitation facilities at the school	No handwashing facilities or no water available at the school

**Table 2 ijerph-20-06914-t002:** Water quality testing results.

School	Water Source and Supply	Free Residual Chlorine (mg/L)	Total Coliforms (MPN/100 mL)	*E. coli* (MPN/100 mL)
1	Municipal: piped water supply	<0.1	<0.1	<1
2	Community-managed: piped water supply	<0.1	<0.1	<1
3	Municipal: piped water supply	<0.1	12.1	3
4	Community-managed: piped water supply	<0.1	22.8	<1
	Community-managed: piped water connected to water filter	<0.1	1	<1
5	Municipal: piped water supply	<0.1	8.6	<1
6	Community-managed: piped water	<0.1	3.1	<1
	Community-managed: protected spring	<0.1	122.3	<1

**Table 3 ijerph-20-06914-t003:** Availability of handwashing stations (HW), soap, and alcohol-based hand rub (ABHR) by school.

School	HW Stations Available at School N	HW Stations with Soap N (%)	ABHR Dispensers Available at School N
1	21	7 (33)	13
2	7	1 (14)	11
3	9	0 (0)	18
4	12	10 (83)	19
5	10	8 (80)	16
6	6	8 (33)	8
Total	65	28 (43)	85

## Data Availability

Data are available from the first author by request.

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
