# Peer review of "Assessment of Water, Sanitation, and Hygiene Conditions in Public Elementary Schools in Quetzaltenango, Guatemala, in the Context of the COVID-19 Pandemic"

_ijerph, 2023, doi:10.3390/ijerph20206914_

Round 1

Reviewer 1 Report

The study offers valuable insights into WASH conditions in the specific context of Guatemala during the COVID-19 pandemic. The work adds to the body of knowledge by providing a baseline assessment and suggesting interventions. The paper assesses the status and functionality of WASH-related services in six public elementary schools in Quetzaltenango, Guatemala, with a focus on hand hygiene, using the WHO/UNICEF Joint Monitoring Programme (JMP) report indicators. The authors report that all schools had water available at the time of the survey, but all water samples at four schools tested positive for total coliforms and at one school, positive for E. coli. Financial constraints and a lack of appropriate WASH infrastructure were the main barriers reported by teachers to meet hand hygiene needs at schools. The authors suggests that appropriate access to WASH infrastructure and supplies could increase hand hygiene practices and improve learning conditions for students. The authors also notes that with the COVID-19 pandemic, the closure of schools posed a risk to the continuity of education and the well-being of children.

Comments to Consider:

The literature review is thorough but could benefit from a broader inclusion of global studies related to WASH in schools. The linkage between previous works and the present study could be further strengthened.

The research design is well laid out, employing appropriate methods for data collection and the selection of schools and the method of assessment are clearly explained. Statistical methods are well-chosen, providing valid results. The results are presented clearly, with well-structured tables and figures. The conclusions are backed by the data, though some additional interpretations could enhance the understanding of the results. Additional insights into the implications of the findings would enhance the manuscript. Important to note that the study was conducted in a specific context (public elementary schools in Quetzaltenango, Guatemala) and the findings may not be generalizable to other settings. Additionally, the study relied on self-reported data from teachers and did not include observations of actual hand hygiene practices.

Language: The manuscript is well-written, and the language is clear and concise.

Grammar: Few minor grammatical errors were noted that do not hinder comprehension. A careful proofreading would be beneficial.

Reviewer 2 Report

I found the paper titled " Assessment of Water, Sanitation, and Hygiene Conditions in Public Elementary Schools in Quetzaltenango, Guatemala, in the context of the COVID-19 pandemic" to be engaging. The authors aim to establish a baseline of WASH services in six public elementary schools in Guatemala, with a focus on hand hygiene.

I am eager to propose some minor revisions for this manuscript. I've outlined a few questions and suggestions below:

  1. Could you elaborate on the novelty of this study or the unique insights it provides in comparison to existing research?
  2. What were the main findings of the study regarding WASH infrastructure in public elementary schools in Guatemala?
  3. How did the researchers assess hand hygiene practices and systems at each school, and what were their results?
  4. Were there any significant differences between schools with regards to access to water, sanitation facilities, or disability-inclusive WASH services?
  5. What are some potential solutions that could address financial constraints and improve access to appropriate WASH infrastructure for these schools?
  6. In light of these findings, how might policymakers prioritize investments in improving WASH services within public elementary schools as a means of reducing transmission of infectious diseases like COVID-19?
  7. The discussion section might benefit from more robust analysis. Integrating additional data from other studies could facilitate meaningful comparisons with the present study's outcomes.
  8. Please provide more data on the importance of physicians and pharmacy around the world to recognize to address the gaps in water, sanitation, and hygiene (WASH) services in public elementary schools.

These revisions should help enhance the manuscript's clarity, depth, and relevance.

 Minor editing of English language required

Reviewer 3 Report

Abstract - line 17: Be more precise about where these samples were taken

Introduction:

Are there any other works published in the literature on the subject of the article? 

I suggest updating and restructuring the introduction with articles with similar themes and showing how your research is innovative.

Line 41: I suggest explaining what these goals

Line 43: which indicators? list some indicators that are relevant to your research

Objectives - lines 57, 58, 59: Improve in all aspects? Or on a few points? I recommend Define more clearly the intention of the study's objective.

Methods

Line 74: explain this methodology

Line 86: reference of WHO recomendations for drinking water quality

Line 88: Were the collections always carried out at the same points? At the same times?

Line 104: Did the interviews follow the same methodology? Or a template of questions? Explain more about the interviews in the different schools. 

Results - 3.2. Water: Availability, quality, and accessibility

No other studies in the same region have identified the non-existence of chlorine in the water distribution system? Or the presence of coliforms in treated water?

3.3. and 3.4

These are important and relevant items that should be discussed in conjunction with the objective of the study.

Line 173 - No other study has identified this fact before?

Reviewer 4 Report

Description:

This study presents assessment of water, sanitation and hygine condions in public elementary schools in Guatemala in the context of the COVID-19 pandemic. Study is written in a legible manner with easy to read. Most chalenges which schools face regarding WASH are mainly related to a lack of sufficient resources and the absence of adequate infrastructre. Data collected in this study will be used to develop interventions and demostrative programs to improve handwashing...

General remarks:

Manuscript seems very interesting, but main lack is Introduction section which needs to be expanded... Introduction section is very poor and doesnot give all the important information. I suggest to expand it and put in it more relevant references related to this study... some informations about COVID-19 related todescribed area, some references to similar studies done in the world...  Introduction part must be written in a sound way... also an overview of the most relevant research that has already been conducted must be given and briefly described...

Specific remarks:

Row 58, 59: Bracket is missing...

Also, some minor editing of English language is required.

Some minor editing of English language is required.
